# Energy Homeostasis Gene Nucleotide Variants and Survival of Hemodialysis Patients—A Genetic Cohort Study

**DOI:** 10.3390/jcm11185477

**Published:** 2022-09-18

**Authors:** Monika Katarzyna Świderska, Adrianna Mostowska, Damian Skrypnik, Paweł Piotr Jagodziński, Paweł Bogdański, Alicja Ewa Grzegorzewska

**Affiliations:** 1Department of Nephrology, Transplantology and Internal Diseases, Poznan University of Medical Sciences, Przybyszewskiego 49, 60-355 Poznań, Poland; 2Department of Diagnostic and Interventional Radiology and Nuclear Medicine, University Medical Center Hamburg-Eppendorf, Martinistraße 52, 20246 Hamburg, Germany; 3Department of Biochemistry and Molecular Biology, Poznan University of Medical Sciences, Święcickiego 6, 60-781 Poznań, Poland; 4Department of Treatment of Obesity, Metabolic Disorders, and Clinical Dietetics, Poznań University of Medical Sciences, Szamarzewskiego 82/84, 60-569 Poznań, Poland

**Keywords:** hemodialysis, cardiovascular mortality, *ANGPTL6* gene variants, DOCK6 gene variants, *FOXO3* gene variants

## Abstract

Background: Patients undergoing hemodialysis (HD) therapy have an increased risk of death compared to the general population. We investigated whether selected single nucleotide variants (SNVs) involved in glucose and lipid metabolism are associated with mortality risk in HD patients. Methods: The study included 805 HD patients tested for 11 SNVs in *FOXO3*, *IGFBP3*, *FABP1*, *PCSK9*, *ANGPTL6*, and *DOCK6* using HRM analysis and TaqMan assays. FOXO3, IGFBP3, L-FABP, PCSK9, ANGPTL6, and ANGPTL8 plasma concentrations were measured by ELISA in 86 individuals. The Kaplan–Meier method and Cox proportional hazards models were used for survival analyses. Results: We found out that the carriers of a C allele in *ANGPTL6* rs8112063 had an increased risk of all-cause, cardiovascular, and cardiac mortality. In addition, the C allele of *DOCK6* rs737337 was associated with all-cause and cardiac mortality. The G allele of *DOCK6* rs17699089 was correlated with the mortality risk of patients initiating HD therapy. The T allele of *FOXO3* rs4946936 was negatively associated with cardiac and cardiovascular mortality in HD patients. We observed no association between the tested proteins’ circulating levels and the survival of HD patients. Conclusions: The *ANGPTL6* rs8112063, *FOXO3* rs4946936, *DOCK6* rs737337, and rs17699089 nucleotide variants are predictors of survival in patients undergoing HD.

## 1. Introduction

Chronic kidney disease (CKD) is a major global health burden, affecting up to 15% of adults [1], with an increase in stage 1 CKD of 15% being observed in the last decade [1]. The most common comorbidities in CKD patients include diabetes mellitus, arterial hypertension, coronary artery disease, peripheral artery disease, anemia, and obesity, which are known mortality risk factors in the general population [1,2,3,4,5,6]. The risk of death in the end-stage renal disease (ESRD) population is four times higher compared to the age and sex-adjusted general population. For example, among individuals aged 40–44 years old, there is more than a 25-year difference in the lifespan between men receiving dialysis and men in the general population and a more than a 30-year difference for women [1].

ESRD is associated with increased levels of oxidative stress and significant abnormalities in circulating lipoproteins and in renal lipid and glucose metabolism [7,8,9]. However, a paradoxical relationship between the lipid profile and obesity and survival has been observed in hemodialysis patients; contrary to the general population, hypercholesterolemia and obesity appear to provide a survival benefit in ESRD patients [10,11]. One possible explanation for both the “cholesterol paradox” and “obesity paradox” is that systemic inflammation and malnutrition are significant confounders of the association between the lipid profile and body mass index (BMI) and mortality in this group of patients [12]. Multiple modifiable and non-modifiable factors are involved in regulating lipid and carbohydrate metabolism, including diet, body composition, levels of physical activity, and genetic factors [13,14]. Therefore, a question arises as to whether the factors associated with lipid and carbohydrate metabolism in general populations would have similar effects on clinical outcomes in hemodialysis patients.

Some studies have shown that nucleotide variants in the insulin growth factor-1 (IGF-1) signaling pathway genes involved in glucose metabolism could influence human longevity [15]. For example, the forkhead box protein O3 gene (*FOXO3,* OMIM*602681) encoding the IGF-1 pathway downstream transcription factor (TF), forkhead box protein O3 (FOXO3), has been found to be strongly associated with human longevity and the prevalence of diabetes and arterial hypertension [16]. Previous in vitro studies found that FOXO3 plays a crucial role in regulating the insulin/insulin-like growth factor 1 (IGF-1)/phosphatidylinositol-3 kinase (PI3K)/AKT (Protein Kinase B) metabolic pathway [17]. Moreover, the transcriptional targets of FOXOs include genes involved in cell cycle arrest, oxidative resistance, apoptosis, autophagy, DNA damage repair, and energy metabolism [18]. Another IGF-1 signaling pathway protein, insulin-like growth factor binding protein 3 (IGFBP-3), the most abundant of the six known IGFBPs, circulates in the bloodstream and transports IGFs that have been sequestered in the form of a ternary complex to the IGF receptors (IGFRs) to initiate a cascade of downstream signaling events [19]. The insulin-like growth factor binding protein-3 gene (*IGFBP3*, OMIM*146732) influences lipid parameters in adolescents and cancer susceptibility in the general population [20,21].

Angiopoietin-like proteins (ANGPTL) are members of a protein family named according to their structural similarity to angiopoietins [22,23,24,25]. Angiopoietin-like protein 6, a liver-derived circulating factor, is considered to be a regulator of metabolic homeostasis [25]. It is encoded by the angiopoietin-like protein 6 gene (*ANGPTL6*, OMIM*609336) and is capable of counteracting both obesity and obesity-related insulin resistance [24]. The production of angiopoietin-like protein 8 (ANGPLT8) in the liver and adipose tissue is induced by insulin via PI3K/AKT signaling [26]. Angiopoietin-like protein 8 gene (*ANGPTL8*) is located in the corresponding intron of dedicator of cytokinesis 6 (*DOCK6,* OMIM*614194) [25].

Moreover, numerous genes associated with lipid metabolism have been found to influence the risk of cardiovascular disease and dyslipidemia in the general population, including the fatty acid binding protein 1 gene (*FABP1,* OMIM*134650). Fatty acid-binding proteins are a family of small and highly conserved cytoplasmic proteins with the ability to bind to long-chain fatty acids and other hydrophobic ligands and participate in the intracellular transportation of lipids [22,27]. 

Proprotein convertase subtilisin/kexin type 9 (PCSK9) is a serine protease that promotes the catabolism of low-density lipoprotein (LDL) receptors (LDLR) and consecutively controls lipid metabolism [28]. Variants of the proprotein convertase subtilisin/kexin type 9 gene (*PCSK9,* OMIM*607786) have recently been associated with cardiovascular risk in the general population [29]. 

Our study aimed to investigate whether the *FOXO3**, IGFBP3, FABP1, PCSK9,* and *ANGPTL6*, *DOCK6* single nucleotide variants and FOXO3, IGFBP3, L-FABP, PCSK9, ANGPTL6, and ANGPTL8 plasma concentrations are associated with the survival probability of Polish hemodialyzed patients

## 2. Materials and Methods

### 2.1. Study Design

The study was designed as a genetic cohort study. STROBE (STrengthening the Reporting of OBservational studies in Epidemiology) guidelines were employed. The study protocol was approved by the Ethics Committee of the Poznań University of Medical Sciences and fulfilled the requirements of the Declaration of Helsinki. The patients were enrolled in the study between June 2016 and December 2017. Mortality was the primary outcome of the study. In deceased individuals, causes of death were registered based on medical documentation and were categorized as cardiac (reported as myocardial infarction, sudden cardiac death, severe arrhythmias, cardiomyopathies, or cardiac failure), vascular (reported as cerebrovascular events, cerebral stroke, or generalized atherosclerosis), infection-related (reported as sepsis, pneumonia, limb necrosis, pyonephrosis, or acute abdomen with peritonitis), cancer-related, and other or unknown. Cardiovascular deaths were defined as deaths caused by cardiac and/or vascular events. Additionally, a subgroup analysis of 83 patients initiating HD therapy was performed. Patients initiating therapy were defined as individuals who had been dialyzed for less than 6 months prior to enrollment in the study. Secondary outcomes were cardiovascular events, diagnosis of coronary artery disease (CAD), BMI, serum total cholesterol, serum high-density lipoprotein cholesterol (HDL-cholesterol), serum low-density cholesterol (LDL-cholesterol), and serum triglycerides (TG). The patient outcomes (death, renal transplantation, moving to a non-collaborating center) were evaluated in January 2020.

### 2.2. Study Participants

Prevalent HD patients (*n* = 805) undergoing dialysis at 27 dialysis centers in the Greater Poland region of Poland were evaluated as candidates for this observational study. All study participants were unrelated Caucasians of Polish origin. The inclusion criteria included written informed consent, age over 18 years, and stable clinical state for at least two months before the onset of the study. The exclusion criteria included missing data in one of the primary or secondary outcomes and secondary causes of hyperlipidemia such as liver disease or a cachectic state at the onset of the study.

The control group included 360 healthy individuals from the same geographical region. The inclusion criteria were written informed consent and age over 18 years. Exclusion criteria encompassed kidney disease (self-reported or as determined by GFR and albumin values), diabetes, cardiovascular disease (CAD, atherosclerosis, or cerebral stroke), and acute infection at the study onset.

### 2.3. Blood Sample Collection

The blood samples were collected from the HD patients before their midweek hemodialysis session and from the healthy volunteers in the morning after an overnight fast. Blood samples were collected in ethylenediaminetetraacetic acid (EDTA) tubes to obtain whole blood and in plasma-separated tubes. After preparation, the blood samples were analyzed or frozen immediately after collection and stored at −80 °C.

### 2.4. Laboratory Methods

Serum concentrations of total cholesterol, HDL-cholesterol, TG, C-reactive protein (CRP), alanine transaminase (ALT), aspartate transaminase (AST), and gamma-glutamyl transferase (GGT) were measured using routine enzymatic methods in a commercial laboratory. The LDL-cholesterol serum concentrations were calculated using the Friedewald formula. In patients with serum TG concentrations ≥ 400 mg/dL, LDL-cholesterol was measured directly (BioSystems S.A., Reagents and Instruments, Barcelona, Spain). 

Plasma concentrations of FOXO3, IGFBP3, L-FABP, PCSK9, ANGPTL6, and ANGPTL8 (betatrophin) were measured in a randomly selected group of 86 HD patients using an enzyme-linked immunosorbent assay (ELISA). Patients were assigned into the group using the simple randomization method. The following commercial kits were used: Human Forkhead Box O3 (FOXO3) ELISA Kit, Catalog Number CSB-E11177h, Cusabio Technology LLC, Wuhan, China; Human IGFBP-3 Immunoassay, Catalog Number DGB300, R&D Systems, Inc., Minneapolis, MN, USA; Human Liver Type Fatty Acid Binding Protein (L-FABP) ELISA Kit, Catalog Number CSB-E13455h, Cusabio Technology LLC, Wuhan, China; Human Proprotein Convertase 9/PCSK9 Immunoassay, Catalog Number DPC900, R&D Systems, Inc., Minneapolis, MN, USA; ANGPTL6 (human) ELISA Kit, Cat. No. AG-45A-0016YEK-KI01, Adipogen Life Sciences, Liestal, Switzerland; and Human Betatrophin ELISA Kit, Catalog No: E11644h, EIAab, Wuhan, China. All laboratory analyzes were performed according to the manufacturer’s instructions. 

### 2.5. Genotyping

DNA was extracted from blood lymphocytes using the salting-out method. Eleven nucleotide variants in six genes were selected based on a literature review. The characteristics of the analyzed SNVs are described in Table 1. Genotyping of the *FOXO3, IGFBP3* rs3110697, *FABP1, PCSK9, ANGPTL6*, and *DOCK6* SNVs was performed using high-resolution melting curve (HRM) analysis with 5x HOT FIREPol EvaGreen HRM Mix (Solis BioDyne, Tartu, Estonia) on the Light Cycler 96 system (Roche Diagnostics, Mannheim, Germany). The primer sequences and conditions for the HRM analyses are presented in Appendix A. The *IGFBP3* rs2854744 variant was analyzed using the predesigned C___1842665_10 TaqMan SNP Genotyping Assay according to the manufacturer’s instructions (Applied Biosystems, Foster City, CA, USA). For quality control, approximately 10% of the randomly chosen samples were regenotyped using the same genotyping method; the concordance rate was 100%. Samples that failed the genotyping analysis were excluded from further statistical analysis. 

### 2.6. Statistical Analyses

All of the statistical analyses were performed using R statistical software [30]. Data are presented as percentages for categorical variables or as medians and ranges for continuous variables that were non-normally distributed. Before every other statistical analysis of quantitative variables, data were checked for normal distribution (Shapiro–Wilk test) and homoscedasticity (Levene’s test). Student’s t-test and ANOVA were used to analyze normally distributed and homoscedastic data sets. Non-normally distributed data sets were compared using Mann–Whitney U, Kruskal–Wallis, and Dunn testing. Spearman’s rank test was used to show correlations between selected variables. Pearson’s chi-squared test and Fisher’s exact test were used to analyze the qualitative variables. The strength of the association was evaluated by odds ratios (ORs) with 95% confidence intervals (95% CI) in a dominant, recessive, and additive model of inheritance. Survival probability since RRT onset was analyzed using the Kaplan–Meier method with the log-rank test. All analyses were performed using patient groups separated by genotypes in three modes of inheritance. The Cox proportional hazards model was applied to show whether and to which extent the effect of a unit change in a covariate was multiplicative concerning the hazard rate (HR) of death. HRs were adjusted for clinical data using Cox proportional hazards regression analysis. Gender, age at RRT onset, myocardial infarction, stroke, diabetic nephropathy, serum concentrations of intact PTH, and calcium phosphate product were applied as clinical variables, possibly contributing to survival probability in the multivariable analyses. A *p*-value of <0.05 was considered to indicate statistical significance. Correction for multiple testing regarding the primary outcome was performed using a 1000-fold permutation version of the k-sample log-rank test. Missing data were not considered in the prevalence calculations and were reported as part of the descriptive statistics. 

The power to detect the genetic associations was determined using the Genetic Association Study (GAS) Power Calculator (http://csg.sph.umich.edu/abecasis/gas_power_calculator/index.html (accessed on 1 March 2022) under the following assumptions: case/control ratio 1.645, significance level = 0.05, prevalence = 63% [9]. It was calculated that a sample size of at least 800 HD patients would yield at least 80% power for detecting the relative risk of 1.25 in the additive and dominant mode of inheritance in all of the analyzed SNVs (Appendix A).

Haplotype frequencies were estimated using Haploview 4.2 software (http://www.broad.mit.edu/mpg/haploview/ (accessed on 2 April 2022). Epistatic interactions between the tested SNVs were analyzed using the odds ratio-based multifactor dimensionality reduction (MDR) method [31]. Statistical significance in both tests was assessed using the 1000-fold permutation test.

## 3. Results

### 3.1. Patient Characteristics

Between June 2016 and December 2017, a group of 1619 prevalent HD patients was screened, and 1205 individuals fulfilled the inclusion criteria and presented no exclusion criteria. In addition, 244 subjects were excluded due to the lack of all of the required measurements. In total, 961 patients were included in the observational study, but 156 were lost to follow-up. The final study group included 805 prevalent HD patients. Table 2 presents the patients’ data. The study group comprised 457 males (56.8%). The median age at RRT onset was 61.4 years. Four hundred thirty-two patients (53.7%) were dialyzed using low-flux dialyzers. All patients were prescribed hemodialysis on a thrice-weekly schedule. During the study period, 510 patients died (63.4%), and 152 underwent renal transplantation (18.9%). Dyslipidaemia, defined according to K/DOQI criteria, was diagnosed in 387 patients (48.1%). Additionally, 337 patients were diagnosed with arterial hypertension (41.8%), 309 patients had coronary artery disease (38.4%), 174 underwent myocardial infarction (21.6%), and 210 suffered from a cerebral stroke (26.1%). The most common cause of the end-stage renal disease was diabetic kidney disease, which was diagnosed in 246 patients, followed by hypertensive nephropathy (*n* = 170), chronic glomerulonephritis (*n* = 105), and chronic tubulointerstitial nephritis (*n* = 67).

### 3.2. Genotype Frequencies in Patients and Controls 

The frequency distributions of the tested SNVs did not differ between the HD patients and the healthy volunteers (Appendix A). Furthermore, all of the SNVs, except for *DOCK6* rs176990893 in the HD patients and *PCSK9* rs11206510 in the controls, complied with the Hardy–Weinberg equilibrium (HWE) (Appendix A). 

### 3.3. Survival Analysis 

Overall, 510 patients died during their RRT periods, which lasted between 0.09 and 29.9 years. Survival probability from the start of RRT was negatively associated with male gender (*p* = 0.013), age at RRT onset (*p* < 0.001), diabetic nephropathy (*p* < 0.001), coronary artery disease (*p* < 0.001), myocardial infarction (*p* < 0.001), cerebral stroke (*p* < 0.001), hypolipemic therapy (*p* = 0.040), body weight (*p* = 0.002), BMI (*p* < 0.001), and plasma concentration of C-reactive protein (*p* < 0.001) and was positively associated with chronic glomerulonephritis as the cause of ESRD (*p* < 0.001), serum concentrations of calcium (*p* = 0.001) and phosphate (*p* = 0.002), intact PTH (*p* < 0.001), and the presence of calcium phosphate product (*p* < 0.001) (Table 3). 

#### 3.3.1. DOCK6 rs737337 and rs17699089 and Mortality in HD Patients

The *DOCK6* rs737337 C allele was associated with all-cause mortality in the recessive mode of inheritance (log-rank test *p* = 0.020; HR 1.94 95% CI 1.09–3.44; Wald test *p* = 0.02; p_corr_ = 0.031) and with cardiac mortality in the additive mode of inheritance (log-rank test *p* = 0.030; HR 2.35 95% CI 1.04–5.32; Wald test *p* = 0.040; p_corr_ = 0.047) (Figure 1). The multivariable regression analysis, which included gender, age at RRT onset, myocardial infarction, stroke, diabetic nephropathy, serum concentrations of intact PTH, and calcium phosphate product, revealed that the *DOCK6* rs737337 C allele remained a significant risk factor for overall and cardiac mortality among clinical factors (HR 2.40 95% CI 1.35–4.28, *p* = 0.003, HR 3.03, 95% CI 1.32–6.95, *p* = 0.009, respectively). The rs737337 C allele was associated with an increased risk of diabetic nephropathy (OR 1.687, 95% CI 1.171–2.432, *p* = 0.005) and a decreased risk of hypertensive nephropathy as a cause of ESRD (OR 0.587, 95% CI 0.363–0-950, *p* = 0.029) in the dominant mode of inheritance (Appendix A). There were no associations between *DOCK6* rs17699089 and survival in the entire group of HD patients.

An additional 3.5-year prospective analysis of 83 patients initiating HD therapy revealed that the *DOCK6* rs737337 C allele was significantly associated with cardiovascular and cardiac mortality in the recessive mode of inheritance (log-rank test *p* < 0.001; HR 68.5 95% CI 4.3–1095.0, Wald test *p* = 0.003, p_corr_ = 0.01 for cardiovascular mortality, and log-rank test *p* < 0.001; HR 68.5 95% CI 4.3–1095.0, Wald test *p* = 0.003, p_corr_ = 0.01 for cardiac mortality) (Figure 2). The G allele of *DOCK6* rs17699089 in the recessive mode of inheritance was associated with an increased risk of all-cause (log-rank test *p* = 0.002; HR 5.7 95% CI 1.6–19.9, Wald test *p* = 0.007, p_corr_ = 0.04), cardiovascular (log-rank test *p* < 0.001; HR 13.9 95% CI 2.6–74.5, Wald test *p* < 0.001, p_corr_ = 0.03), and cardiac mortality (log-rank test *p* < 0.001; HR 20.3 95% CI 3.2–127.9, Wald test *p* < 0.001, p_corr_ = 0.02) (Figure 2).

#### 3.3.2. ANGPTL6 rs8112063

In the unadjusted analyses, the C allele of *ANGPTL6* rs8112063 in the dominant mode of inheritance was associated with increased all-cause (log-rank test *p* = 0.047; HR 1.21 95% CI 1.00–1.48; Wald test *p* = 0.050; p_corr_ = 0.048), cardiovascular (log-rank test *p* = 0.010; HR 1.39 95% CI 1.07–1.81; Wald test *p* = 0.010; p_corr_ = 0.017), and cardiac mortality (log-rank test *p* = 0.003; HR 1.64 95% CI 1.18–2.27; Wald test *p* = 0.003; p_corr_ = 0.004) (Figure 3). The *ANGPTL6* rs8112063 C allele was not an independent risk factor for overall mortality after adjustment for gender, age at RRT onset, myocardial infarction, stroke, diabetic nephropathy, serum concentrations of intact PTH, and calcium phosphate product (HR 1.14, 95% CI 0.94–1.39, *p* = 0.192), but it remained a significant risk factor for cardiac and cardiovascular mortality after adjustment (HR 1.59, 95% CI 1.15–2.21, *p* = 0.005, HR 1.31, 95% CI 1.01–1.72, *p* = 0.042, respectively). The C allele of *ANGPTL6* rs8112063 was also associated with an increased risk of diabetic nephropathy (OR 1.503, 95% CI 1.054–2.143, *p* = 0.024) and a decreased risk of tubulointerstitial nephropathy as a cause of ESRD in the recessive mode of inheritance (OR 0.349, 95% CI 0.148–0.825, *p* = 0.012). Patients with the *ANGPTL6* rs8112063 CC genotype were less likely to be diagnosed with dyslipidemia by K/DOQI than the bearers of the TT genotype (OR 0.672, 95% CI 0.453–0.997, *p* = 0.048). Carriers of the rs8112063 C allele had lower serum concentrations of LDL-cholesterol and higher concentrations of HDL-cholesterol than those with the TT genotype (*p* = 0.014 and *p* = 0.026 in the dominant mode of inheritance) (Appendix A).

#### 3.3.3. FOXO3 rs4946936

The minor T allele of FOXO3 rs4946936 was associated with a decreased risk of cardiac (log-rank test *p* = 0.040; HR 0.57 95% CI 0.33–0.98; Wald test *p* = 0.040; p_corr_ = 0.042) and cardiovascular death (log-rank test *p* = 0.04; HR 0.64 95% CI 0.42–0.99; Wald test *p* = 0.04; p_corr_ = 0.047) in HD patients (Figure 4). FOXO3 rs4946936 was not associated with cardiac and cardiovascular mortality after adjustments for gender, age at RRT onset, myocardial infarction, stroke, diabetic nephropathy, serum concentrations of intact PTH, and calcium phosphate product (HR 0.64, 95% CI 0.37–1.11, *p* = 0.109 for cardiac mortality and HR 0.75, 95% CI 0.48–1.16, *p* = 0.191 for cardiovascular mortality). Patients with the FOXO3 rs4946936 T allele were less likely to be diagnosed with dyslipidemia according to the K/DOQI criteria (OR 0.686, 95% CI 0.517–0.910, *p* = 0.009) and had lower serum concentrations of total cholesterol and LDL-cholesterol than the bearers of the CC genotype (*p* = 0.029 and 0.036, respectively) (Appendix A).

#### 3.3.4. Other Genotypes

There were no associations between the risk of death in HD patients and the other analyzed SNVs (*FOXO3* rs2802292; *IGF2BP2* rs4402960, and rs11705701; *IGFBP3* rs3110697, and rs2854744; *FABP1* rs2241883, and rs2919872; *PCSK9* rs562556, and rs11206510; and *DOCK6* rs17699089) (Appendix A).

### 3.4. Protein Plasma Concentrations

There were no statistically significant differences in the genotype distributions of the tested nucleotide variants among the patients in which the plasma concentrations of the tested proteins were evaluated and in those in which they were not measured, except for *ANGPTL6* rs8112063. The study participants with measured protein ANGPTL6 concentrations had a lower frequency distribution of the rare rs8112063 C allele than the other group (*p* = 0.021) (Appendix A). There were no associations between the plasma concentrations of FOXO3, IGFBP3, L-FABP, PCSK9, ANGPTL6, and ANGPTL8 and survival probability in a 3.5-year prospective analysis from June 2016 (Appendix A). Patients with the CC genotype of *FABP1* rs2241883 had higher L-FABP plasma concentrations than the carriers of the dominant T allele (57.9, 13.5–85.7 vs. 32.5 ng/dL, 4.0–114.3 ng/dL, *p* = 0.023). Bearers of the CC genotype of *FABP1* rs2919872 had lower FABP1 plasma concentrations than those with the T allele (22.4, 9.9–88.8 vs. 41.7 ng/dL, 4.0–114.3 ng/dL, *p* = 0.037) (Appendix A). There were no correlations between the other analyzed nucleotide variants and the serum concentrations of their protein products (Appendix A).

### 3.5. Haplotype Analysis

The *DOCK6* rs17699089G_rs737337C haplotype was positively associated with diabetic nephropathy. There were no other significant associations between the *DOCK6* haplotypes and the tested phenotypes (Table 4).

### 3.6. Epistatic Interactions

A gene–gene interaction was noted among *FOXO3* rs4946936 and *ANGPTL6* rs8112063 (testing balance accuracy (TBA) 0.58, *p*-value 0.002) as well as among the *FOXO3* rs4946936, *IGFBP3* rs2854744, *FABP1* rs2919872, and *ANGPTL6* rs8112063 (TBA 0.56, *p*-value 0.027) and *FOXO3* rs4946936, *IGFBP3* rs2854744, *FABP1* rs2241883, *FABP1* rs2919872, and *ANGPTL6* rs8112063 nucleotide variants in relation to all-cause mortality (TBA 0.55, *p*-value 0.026, respectively) (Table 5). Moreover, an epistatic interaction between *FOXO3* rs2802292, *IGFBP3* rs2854744, *FABP1* rs2241883, and *ANGPTL6* rs8112063 was observed in relation to cardiovascular mortality (TBA 0.53, *p*-value 0.024) (Table 6). *IGFBP3* rs3110697, *DOCK6* rs737337, and *DOCK6* rs17699089 showed a gene–gene interaction concerning the diagnosis of myocardial infarction (TBA 0.63, *p*-value 0.032) (Table 7). Furthermore, there was a gene–gene interaction between *FOXO3* rs2802292 and *FABP1* rs2241883 regarding the diagnosis of dyslipidemia by K/DOQI (TBA 0.59, *p*-value 0.021) (Appendix A). No further significant gene–gene interactions were detected in relation to other comorbidities, including diabetes (Appendix A).

## 4. Discussion

We conducted a genetic cohort study to assess whether nucleotide variants of selected genes associated with glucose and lipid metabolism are associated with mortality risk in patients undergoing hemodialysis treatment. We first demonstrated that the *ANGPTL6* rs8112063 and *FOXO3* rs4946936 as well as the *DOCK6* rs737337 and rs17699089 SNVs are predictors of survival in Polish HD patients. In addition, we also noted a gene–gene interaction between *FOXO3* rs4946936 and *ANGPTL6* rs8112063 and between *FOXO3* rs4946936, *IGFBP3* rs2854744, *FABP1* rs2241883, *FABP1* rs2919872, and *ANGPTL6* rs8112063 regarding overall survival. To our knowledge, this study is the first to show that the *FOXO3*, *ANGPTL6*, and *DOCK6* variants can predict clinical outcomes in the uremic population.

End-stage renal disease patients have poorer survival than the general population [1]. Mortality is exceptionally high within the first few months of dialysis [32]. Cardiovascular diseases remain the most common cause of death among RRT patients, accounting for almost 50% of mortality [33]. In our study, 63% of patients died over the course of 3.5 years, and cardiovascular diseases were the most common cause of death. The univariate analyses confirmed multiple known demographic, clinical, and laboratory mortality risk factors of HD patients, such as male gender, advanced age, diabetes, history of cardiovascular diseases, and CRP concentrations [34,35,36,37,38]. We did not observe a significant relationship between the lipid profile and survival. Patients on hypolipemic therapy had a slightly increased risk of death compared to those not taking lipid-lowering drugs. This paradoxical association might be confounded by a greater prevalence of comorbidities among patients receiving hypolipemic therapy [39]. Unlike previous studies, we observed no associations between smoking status and overall survival [2,40]. Contrary to many studies, BMI and body weight were negatively associated with survival probability in our study [2,41].

Angiopoietin-like proteins, a family of proteins that regulate energy and glucose homeostasis, are involved in angiogenesis and share similarities with angiopoietins, such as a coiled-coil domain, a linker region, and a carboxy-terminal fibrinogen-like domain [42,43]. Circulating levels of several proteins in the ANGPTL family have been associated with obesity, diabetes, and mortality in HD patients [44,45].

ANGPTL6 has been identified as a circulating mediator of angiogenesis that is capable of increasing endothelial permeability [42]. Previous studies revealed increased serum concentrations of this anti-obesity hepatokine in individuals with metabolic syndrome and type 2 diabetes and decreased circulating ANGPTL6 levels in hemodialyzed individuals compared to healthy subjects [44,46,47]. rs8112063 is located in the first intron of *ANGPTL6*. As the introns (especially the first one) are known to be regulatory regions that modulate gene expression, rs8112063 could potentially influence ANGPTL6 expression levels [24]. Our study revealed that the rs8112063 C allele was a significant risk factor of all-cause, cardiovascular, and cardiac mortality in HD patients, and it remained an independent cardiovascular mortality risk factor after adjustment for gender, age at RRT onset, myocardial infarction, stroke, diabetic nephropathy, and serum concentrations of intact PTH and calcium phosphate product. Hostettler et al. observed that several rare *ANGPTL6* genetic variants are risk factors for intracranial aneurysms [48]. However, we did not observe an association between the tested *ANGPTL6* nucleotide variant HD and vascular mortality. Interestingly, the French MONICA study suggested that the C allele of rs8112063 could be a risk factor for metabolic syndrome [24]. We found that the carriers of the rs8112063 CC genotype were more likely to suffer from diabetic nephropathy and less likely to be diagnosed with dyslipidemia by K/DOQI compared to those with the T allele. To our knowledge, this is the first study to report such an association. Interestingly, mice knockout studies have shown that ANGPTL6 counteracts diet-induced obesity and insulin resistance via increasing energy expenditure, which suggests that ANGPTL6 may have a modulatory role in diabetes [49]. However, patients with type 2 diabetes have paradoxically increased circulating ANGPTL6 concentrations compared to healthy individuals [47]. ANGPTL6 could therefore contribute to the development of diabetic nephropathy through pathways related to insulin resistance, which is associated with greater salt sensitivity, increased glomerular pressure, albuminuria, and kidney function decline [50].

*ANGPLT8*, a gene located in the corresponding intron of *DOCK6,* encodes ANGPTL8, a peptide hormone produced in the liver and adipose tissue that plays a role in glucose and lipid homeostasis [25]. rs737337 is located 2.8 kb upstream of the *ANGPTL8* transcription start site and is a synonymous variant in exon 19 of *DOCK6* (Thr723). In a study by Cannon et al. rs737337 showed strong enhancer activity in transcriptional reporter assays [51]. Our study found that the C allele of *DOCK6* rs737337 was a risk factor for all-cause and cardiac mortality among HD patients. Notably, *DOCK6* rs737337 remained a cardiac mortality risk factor after adjusting for the clinical parameters that are relevant to survival. In contrast, Agra et al. found that the C allele of *DOCK6* rs737337 was associated with a better prognosis in obese patients with heart failure [52]. Interestingly, Zou et al. found that serum ANGPTL8 levels were associated with the risk of all-cause mortality in diabetic subjects [53]. Furthermore, we observed that the C allele of rs737337 was associated with an increased risk of diabetic nephropathy as a cause of ESRD. 

Another tested *DOCK6* variant, rs17699089, was not directly associated with mortality or diabetic nephropathy in the entire group of Polish HD subjects. However, in a prospective analysis of patients initiating HD therapy, we observed an association between the G allele of rs17699089 and all-cause, cardiovascular, and cardiac mortality. Rs17699089, a *DOCK6* intron variant, showed evidence of allelic differences in the transcriptional activity of *ANGPTL8* in subcutaneous adipose tissue [51]. Moreover, a study by Ghasemi et al. found that rs17699089 was in linkage disequilibrium with rs2278426, a known *ANGPTL8* variant associated with serum concentrations of total cholesterol and an increased risk of diabetic nephropathy [51,54]. Interestingly, we found a significant haplotype association between *DOCK6* rs17699089_rs737337 and diabetic nephropathy. To date, there have been no studies on the putative associations between DOCK6 and diabetes or CKD. However, multiple studies have revealed that the levels of ANGPTL8 are higher in patients with diabetic nephropathy and obesity as well as positively correlated with atherogenic markers and carotid intima-media thickness [55,56,57,58]. Another study by Zou et al. revealed that older subjects with higher circulating ANGPTL8 levels were at an increased risk of kidney function decline, which might suggest a role of ANGPTL8 in the pathogenesis of CKD [59]. ANGPTL8 may drive the progression of diabetic nephropathy through pathways and mechanisms related to insulin resistance or through inflammatory mechanisms [60,61]. However, we observed no differences in the frequency distributions of *DOCK6* SNVs among the HD patients and controls. In our study, *DOCK6* rs737337 and rs17699089 interacted with *IGFBP3* rs3110697 in relation to myocardial infarction diagnosis among HD patients. Interestingly, both rs17699089 and rs737337 are associated with HDL-cholesterol levels in the general population [51]. Moreover, GWAS analyses revealed that rs737337 is associated with the lipid profile and CAD susceptibility in the European population [62,63]. 

The forkhead box (FOX) is a heterogenic protein family of transcription factors that contain a conserved DNA-binding domain, a sequence of 80 to 100 amino acids called the forkhead domain [64]. FOXO3 plays a vital role in multiple fundamental processes in cells, such as controlling metabolism, cell division and differentiation status, and response to cellular stress [65]. *FOXO3* nucleotide variants, especially rs2802292, exhibit a consistently replicated genetic association with longevity in multiple populations worldwide [66]. rs4946936 is located in the 3′UTR of *FOXO3*. Wang et al. found that the rs4946936 T allele significantly increased the expression levels in luciferase assays and affected the binding affinity of miR-223 to the *FOXO3* 3′UTR [67]. Our analysis revealed that the T allele of *FOXO3* rs4946936 was a protective factor against cardiovascular and cardiac mortality. There was also a gene–gene interaction between *FOXO3* rs4946936 and *ANGPTL6* rs8112063 regarding all-cause mortality. This observation is concordant with the results of earlier studies on centenarians, which also reported protective effects of the rs4946936 T allele [68,69]. Surprisingly, there were no direct associations between *FOXO3* rs2802292 and survival in HD patients. However, we observed an epistatic interaction between *FOXO3* rs2802292, *IGFBP3* rs2854744, *FABP1* rs2241883, and *ANGPTL6* rs8112063 and cardiovascular mortality in Polish HD subjects. Associations between *FOXO3* rs2802292 and the prevalence of essential hypertension and improved metabolic control in diabetic patients have been shown in previous studies [70,71]. On the other hand, Klinpudtan et al. reported that the longevity-associated G allele of *FOXO3* rs2802292 appears to have contrasting associations with heart disease prevalence according to sex in older Japanese people [16]. The FOXO3 protein might play a protective role in chronic kidney disease by preventing mitochondrial damage and ameliorating fibrosis [72,73]. However, we did not note any differences in the frequency distributions of FOXO3-related genetic alterations among HD patients and controls.

Our study has several limitations. Firstly, the primary survival analysis was evaluated from the onset of RRT therapy, which makes it more prone to selection bias than other epidemiologic studies. To address this problem, we adjusted the survival analyses for selected clinical variables associated with death risk in HD patients, evaluated the relationship between the tested SNVs and other clinical phenotypes, and conducted an additional survival analysis among the HD patients who had started RRT less than 6 months prior to study enrollment. Moreover, despite a decent study population of over 800 HD patients, there were only 55 and 53 cases of death due to neoplasms and infections, respectively. Because of that, a dedicated cause-specific survival analysis of infectious or neoplastic deaths was not conducted to avoid underpowered statistical tests and inconclusive results. Thirdly, the investigation was performed on a specific population comprising Polish dialyzed patients, and the findings should be used with caution when extrapolated to other ethnicities. Further investigations of putative genetic predictors of survival in larger cohorts encompassing diverse ethnicities are necessary. The results of our study should also be confirmed in other cohorts of Caucasian patients. Finally, plasma concentrations of protein products of the tested nucleotide variants were only evaluated in a limited number of subjects. Nevertheless, the ELISA analysis of the tested proteins was performed in randomly chosen patients to increase the credibility of the results. However, the group in which plasma protein concentrations were measured had lower frequency distributions of the rare C allele of *ANGPTL6* rs8112063 compared to the other study participants, which could have potentially led to underpowered tests. Future investigations with larger study sizes and examining the tested protein concentrations over extended periods are needed to elucidate their role in ESRD. 

## 5. Conclusions

In summary, our study demonstrated that *ANGPTL6* rs8112063 and *DOCK6* rs737337 SNVs are significant predictors of all-cause and cardiac mortality in Polish HD patients. In addition, the rare allele of *DOCK6* rs17699089 is associated with all-cause, cardiovascular, and cardiac mortality in patients initiating HD therapy. Moreover, the rare variant of *FOXO3* rs4946936 is a protective factor against cardiac and cardiovascular death in this group. However, further prospective studies with larger study sizes are needed to clarify the influence of the genes associated with glucose and lipid metabolism on the clinical outcomes of HD patients.

## Figures and Tables

**Figure 1 jcm-11-05477-f001:**
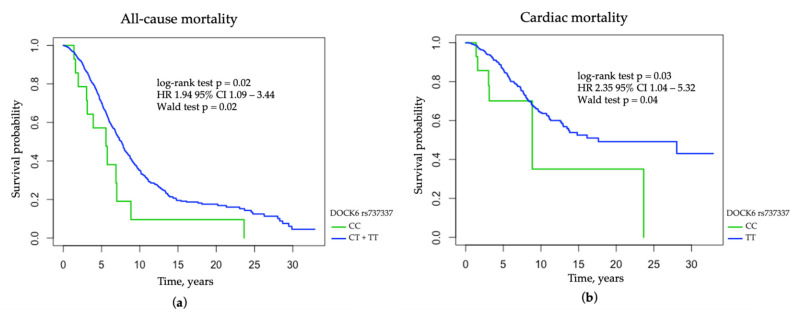
The probability of survival in hemodialysis patients concerning *DOCK6* rs737337 variant: (**a**) all-cause mortality among HD patients concerning *DOCK6* rs737337 variant in the recessive mode of inheritance; (**b**) cardiac mortality among HD patients concerning *DOCK6* rs737337 variant in the additive mode of inheritance.

**Figure 2 jcm-11-05477-f002:**
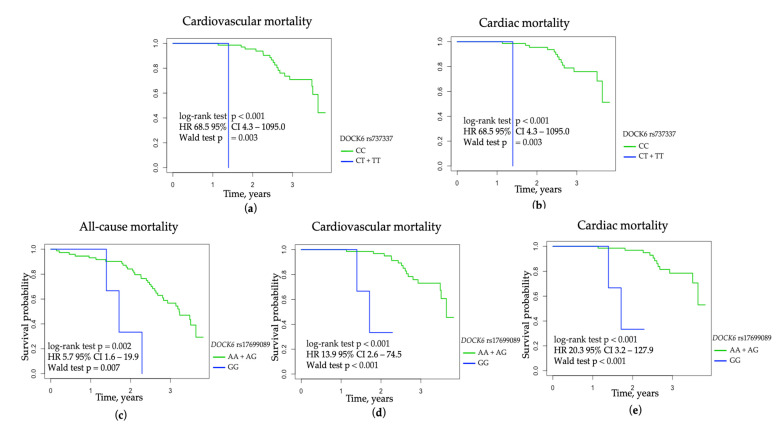
The probability of survival in 83 patients initiating HD therapy concerning *DOCK6* variants: (**a**) cardiovascular mortality among patients initiating HD therapy concerning *DOCK6* rs737337 variant in the recessive mode of inheritance; (**b**) cardiac mortality among patients initiating HD therapy concerning *DOCK6* rs737337 variant in the recessive mode of inheritance; (**c**) all-cause mortality among patients initiating HD therapy concerning *DOCK6* rs17699089 in the recessive mode of inheritance; (**d**) cardiovascular mortality among patients initiating HD therapy concerning *DOCK6* rs17699089 in the recessive mode of inheritance; (**e**) cardiac mortality among patients initiating HD therapy concerning *DOCK6* rs17699089 in the recessive mode of inheritance.

**Figure 3 jcm-11-05477-f003:**
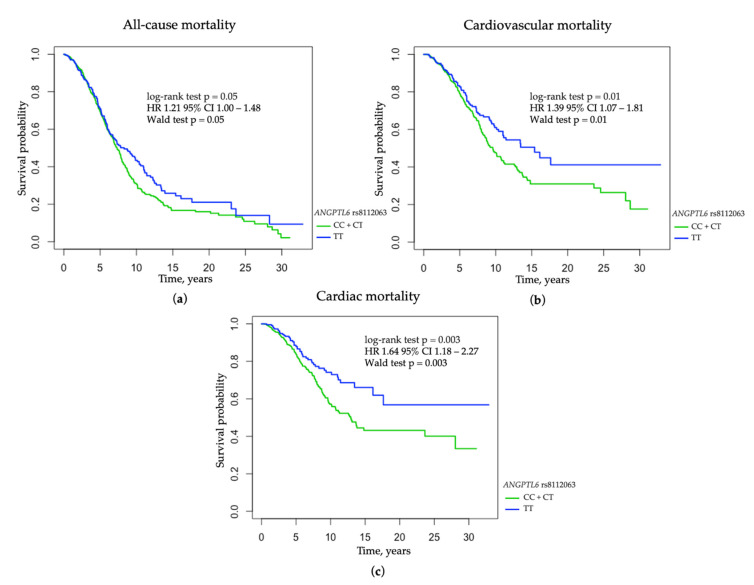
The probability of survival in hemodialysis patients concerning *ANGPTL6* rs8112063 variant: (**a**) all-cause mortality among HD patients concerning *ANGPTL6* rs8112063 variant in the dominant mode of inheritance; (**b**) cardiovascular mortality among HD patients concerning *ANGPTL6* rs8112063 variant in the dominant mode of inheritance; (**c**) cardiac mortality among HD patients with respect to *ANGPTL6* rs8112063 variant in the dominant mode of inheritance.

**Figure 4 jcm-11-05477-f004:**
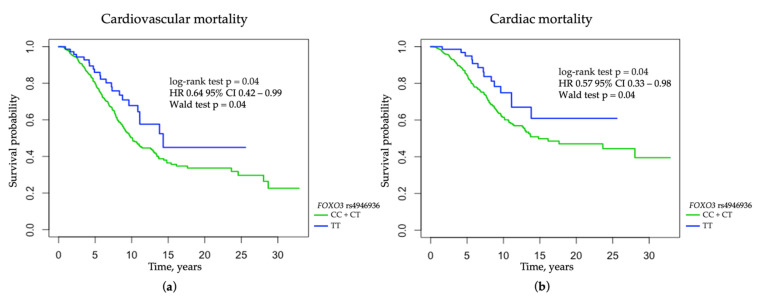
The probability of survival in hemodialysis patients with respect to FOXO3 rs4946936 variant: (**a**) cardiovascular mortality among HD patients with respect to FOXO3 rs4946936 variant in the recessive mode of inheritance; (**b**) cardiac mortality among HD patients with respect to FOXO3 rs4946936 variant in the recessive mode of inheritance.

**Table 1 jcm-11-05477-t001:** Characteristics of the analyzed nucleotide variants.

Gene Symbol	rs No.	Location ^1^	SNV Function ^2^	Alleles ^3^	MAF ^4^	MAF ^5^
*FOXO3*	rs2802292	chr6:108587315	Intron	G/T	0.433	0.378
*FOXO3*	rs4946936	chr6:108682118	3 Prime UTR	C/T	0.345	0.297
*IGFBP3*	rs3110697	chr7:45915430	Intron	A/G	0.429	0.419
*IGFBP3*	rs2854744	chr7:45921476	2KB Upstream	G/T	0.457	0.460
*FABP1*	rs2241883	chr2:88124547	Missense	C/T	0.348	0.325
*FABP1*	rs2919872	chr2:88129052	2KB Upstream	C/T	0.466	0.448
*PCSK9*	rs562556	chr1:55058564	Missense	A/G	0.179	0.172
*PCSK9*	rs11206510	chr1:55030366	-	C/T	0.172	0.184
*ANGPTL6*	rs8112063	chr19:10099035	Intron	C/T	0.434	0.424
*DOCK6*	rs737337	chr19:11236817	Synonymous	C/T	0.073	0.080
*DOCK* *6*	rs17699089	chr19:11233119	Intron	A/G	0.089	0.108

^1^ NCBI build 38/hg38. ^2^ According to the Single Nucleotide Polymorphism database (dbSNP). ^3^ Underline denotes the minor allele. ^4^ MAF, minor allele frequency (1000 Genomes project, EUR samples). ^5^ MAF, minor allele frequency (ALFA Allele Frequency, EUR samples). Abbreviations: *ANGPTL6:* angiopoietin-like protein 6; *DOCK6*: dedicator of cytokinesis 6; *FABP1:* fatty acid-binding protein 1; *FOXO3*: forkhead box protein O3; *IGFBP3*: insulin growth factor binding protein 3; *PCSK9*: proprotein convertase subtilisin/kexin type 9.

**Table 2 jcm-11-05477-t002:** Demographic, clinical, and laboratory data of HD patients, *n* = 805.

Parameter	Value
Clinical data	
Male gender, *n* (%)	457 (56.8%)
Age, years	67.9 (18–95.9)
Age at RRT onset, years	61.4 (8.7–91.7)
RRT vintage, years	5.8 (0.04–32.9)
LF-HD, *n* (%)	432 (53.7%)
HF-HD, *n* (%)	321 (39.9%)
HDF, *n* (%)	52 (6.5%)
Beginning of RRT with peritoneal dialysis, *n* (%)	20 (2.5%)
Transplantation, *n* (%)	152 (18.9%)
Diabetic kidney disease as the cause of ESRD, *n* (%)	246 (30.6%)
Hypertensive nephropathy as the cause of ESRD, *n* (%)	170 (21.1%)
Chronic glomerulonephritis as the cause of ESRD, *n* (%)	105 (13.0%)
Chronic tubulointerstitial nephritis as the cause of ESRD, *n* (%)	67 (8.3%)
Coronary artery disease, *n* (%)	309 (38.4%)
Myocardial infarction, *n* (%)	174 (21.6%)
Cerebral stroke, *n* (%)	210 (26.1%)
Dyslipidemia by K/DOQI, *n* (%)	387 (48.1%)
Hypolipemic therapy, *n* (%)	334 (41.5%)
Arterial hypertension, *n* (%)	337 (41.8%)
Smoker, *n* (%)	132 (16.4%)
Weight, kg	73.1 (31–196)
Height, m	1.68 (1.28–1.93)
BMI, kg/m^2^	25.7 (14.3–59.2)
Causes of death	
All, *n* (%)	510 (63.4%)
Cardiovascular, *n* (%)	312 (38.8%)
Cardiac, *n* (%)	224 (27.8%)
Vascular, *n* (%)	88 (10.9%)
Neoplasms, *n* (%)	55 (6.8%)
Infectious, *n* (%)	53 (6.6%)
Laboratory data	
AST, U/L	15.0 (3.0–177.0)
ALT, U/L	14.0 (0.6–195.0)
GGT, U/L	30.0 (1.0–682.0)
ALP, U/L	97.0 (12.3–1408.0)
Intact PTH, pg/mL	389.1 (5.5–3757.0)
Ca, mg/dL	8.8 (6.0–12.8)
Phosphate, mg/dL	5.1 (1.8–11.3)
Calcium phosphate product	44.6 (16.7–108.7)
CRP, mg/L	5.6 (0.1–195.0)
Total cholesterol, mg/dL	171.0 (72.0–626.0)
HDL-cholesterol, mg/dL	40.0 (6.0–118.0)
LDL-cholesterol, mg/dL	95.6 (25.0–512.0)
TG, mg/dL	149.0 (26.0–856.0)
Non-HDL-cholesterol, mg/dL	129.0 (8.0–593.0)
Triglyceride to HDL-cholesterol, mg/dL	3.79 (0.43–49.71)

Abbreviations: ALP: alkaline phosphatase; ALT: alanine aminotransferase; AST: aspartate aminotransferase; BMI: body mass index; CRP: C-reactive protein; ESRD: end-stage renal disease; GGT: gamma-glutamyl transferase; HD: hemodialysis; HDF: hemodiafiltration; HDL: high-density lipoprotein; HF: high flow; K/DOQI: Kidney Disease Outcomes Quality Initiative; LDL: low-density lipoprotein LF: low flow; PTH: parathormone; RRT: renal replacement therapy; TG: triglycerides.

**Table 3 jcm-11-05477-t003:** Clinical variables associated with overall survival in HD patients, *n* = 805.

Parameter	Odds Ratio (95% CI)	*p*-Value ^1^
Male gender	1.250 (1.048–1.491)	0.013
Age at RRT onset, years	1.050 (1.042–1.057)	<0.001
Diabetic nephropathy	1.770 (1.467–2.135)	<0.001
HF-HD/HDF vs. LF-HD	1.006 (0.844–1.199)	0.947
Hypertensive nephropathy	1.163 (0.942, 1.435)	0.161
Chronic glomerulonephritis	0.446 (0.333–0.599)	<0.001
Chronic tubulointerstitial nephritis	0.790 (0.576–1.084)	0.144
Coronary artery disease	1.728 (1.452–2.058)	<0.001
Myocardial infarction	1.725 (1.420–2.095)	<0.001
Cerebral stroke	1.405 (1.163–1.698)	<0.001
Dyslipidemia by K/DOQI criteria	0.910 (0.764–1.083)	0.289
Hypolipemic therapy	1.201 (1.008–1.431)	0.040
Smoking	1.042 (0.816–1.330)	0.741
Body weight (kg)	1.009 (1.005–1.014)	0.002
Height (m)	1.843 (0.762–4.459)	0.175
BMI (kg/m^2^)	1.034 (1.017–1.051)	<0.001
CRP (mg/L)	1.008 (1.004–1.011)	<0.001
Total cholesterol (mg/dL)	0.999 (0.998–1.001)	0.504
LDL-cholesterol (mg/dL)	0.999 (0.997–1.002)	0.591
HDL-cholesterol (mg/dL)	0.995 (0.988–1.002)	0.156
TG (mg/dL)	1.000 (0.999–1.001)	0.882
Ca (mg/dL)	0.834 (0.747–0.932)	0.001
Phosphate (mg/dL)	0.911 (0.859–0.968)	0.002
Intact PTH (100 pg/mL)	0.967 (0.949–0.986)	<0.001
Calcium phosphate product (mg^2^/dL^2^)	0.988 (0.981–0.994)	<0.001

^1^ Wald test statistics using Cox proportional hazards model.

**Table 4 jcm-11-05477-t004:** Associations between *DOCK6* rs17699089_rs737337 haplotypes and selected clinical parameters.

Parameter	Haplotype	Case, Control Frequencies	*p*-Value	*p*_corr_ Value ^1^	OR (95%CI) ^2^, *p*-Value	OR (95%CI) ^3^, *p*-Value
Overall mortality	AT	0.817, 0.858	0.037	0.079	0.74 (0.56–0.98), 0.037	Reference
	GC	0.119, 0.085	0.032	0.066	1.46 (1.03–2.64), 0.033	1.47 (1.04–2.09), 0.029
	GT	0.062, 0.057	0.725	0.990	1.08 (0.70–1.67), 0.719	1.13 (0.73–1.75), 0.578
Cardiac mortality	AT	0.807, 0.841	0.104	0.240	0.79 (0.60–1.05), 0.104	Reference
	GC	0.131, 0.097	0.054	0.171	1.39 (0.99–1.95), 0.054	1.40 (1.00–1.96), 0.052
	GT	0.060, 0.060	0.947	1.000	0.99 (0.62–1.56), 0.951	1.03 (0.65–1.63), 0.907
Cardiovascular mortality	AT	0.833, 0.831	0.932	1.000	1.01 (0.77–1.32), 0.932	Reference
	GC	0.117, 0.100	0.292	0.688	1.17 (0.85–1.62), 0.333	1.15 (0.83–1.59), 0.402
	GT	0.049, 0.068	0.119	0.338	0.70 (0.45–1.10), 0.120	0.72 (0.46–1.12), 0.142
Diabetic nephropathy	AT	0.804, 0.844	0.051	0.199	0.76 (0.58–1.00), 0.051	Reference
	GC	0.138, 0.093	0.008	0.028	1.56 (1.12–2.16), 0.008	1.55 (1.12–2.16), 0.008
	GT	0.056, 0.062	0.627	1.000	0.89 (0.56–1.41), 0.620	0.94 (0.59–1.49), 0.795
CAD	AT	0.835, 0.830	0.803	1.000	1.04 (0.79–1.36), 0.800	Reference
	GC	0.120, 0.098	0.181	0.404	1.24 (0.90–1.72), 0.182	1.21 (0.87–1.67), 0.250
	GT	0.044, 0.070	0.029	0.056	0.60 (0.38–0.95), 0.029	0.62 (0.39–0.98), 0.038

^1^*p*-value calculated using permutation test and a total of 1000 permutations. ^2^ All other haplotypes were pooled together and used as the reference. ^3^ The most common haplotype was used as the reference. Abbreviations: CAD: coronary artery disease.

**Table 5 jcm-11-05477-t005:** Epistatic interactions between the analyzed genes in HD patients who died and those who survived in a survival analysis.

No. of Risk Variants	Models	Testing Balanced Accuracy	CVC	OR-MDR	95%CI	*p*-Value ^1^
2	*FOXO3* rs4946936_*ANGPT**L**6* rs8112063	0.58	4/10	0.557	(0.820–1.699)	0.002
3	*FOXO3* rs2802292_*FABP1* rs2241883_*ANGPTL6* rs8112063	0.55	3/10	0.349	(0.084–1.451)	0.127
4	*FOXO3* rs4946936_*IGFBP3* rs2854744_*FABP1* rs2919872_*ANGPTL6* rs8112063	0.56	8/10	0.116	(0.014–0.992)	0.027
5	*FOXO3* rs4946936_IGFBP3 rs2854744_FABP1 rs2241883_*FABP1* rs2919872_*ANGPTL6* rs8112063	0.55	5/10	0.116	(0.014–0.992)	0.026
8	*FOXO3* rs2802292_ *FOXO3* rs4946936_*IGFBP3* rs3110697_*IGFBP3* rs2854744_*FABP1* rs2241883_*FABP1* rs2919872_*PCSK9* rs562556_*ANGPTL6* rs8112063	0.59	4/10	0.291	(0.027–3.197)	0.306

Abbreviations: CVC, cross-validation consistency; OR-MDR, odds ratio-multifactor-dimensionality reduction; 95% CI, confidence interval 95%. ^1^ Significance of accuracy, empirical *p*-value based on 1000 permutations.

**Table 6 jcm-11-05477-t006:** Epistatic interactions between the analyzed genes with respect to cardiovascular mortality in HD patients.

No. of Risk Variants	Models	Testing Balanced Accuracy	CVC	OR-MDR	95%CI	*p*-Value ^1^
2	*PCSK9 rs562556*_*DOCK6**rs737337*	0.59	6/10	2.212	(1.127–4.341)	0.993
3	*FABP1 rs2919872*_*ANGPTL6 rs8112063*_*DOCK6 rs737337*	0.57	4/10	0.453	(0.170–1.206)	0.074
4	*FOXO3 rs2802292*_*IGFBP3 rs2854744*_*FABP1 rs2241883*_*ANGPTL6 rs8112063*	0.53	4/10	0.148	(0.019–1.142)	0.024

^1^ Significance of accuracy, empirical *p*-value based on 1000 permutations.

**Table 7 jcm-11-05477-t007:** Epistatic interactions between the analyzed genes with respect to myocardial infarction in HD patients.

No. of Risk Variants	Models	Testing Balanced Accuracy	CVC	OR-MDR	95%CI	*p*-Value ^1^
2	*IGFBP3 rs3110697*_ *PCSK9 rs11206510*	0.68	3/10	0.603	(0.238–1.525)	0.191
3	*IGFBP3 rs3110697*_ *DOCK6**rs737337*_ *DOCK6 rs17699089*	0.63	3/10	0.383	(0.139–1.056)	0.032

^1^ Significance of accuracy, empirical *p*-value based on 1000 permutations.

## Data Availability

The de-identified datasets generated through this study can be provided by the corresponding author upon request.

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
