# Peer review of "Energy Homeostasis Gene Nucleotide Variants and Survival of Hemodialysis Patients—A Genetic Cohort Study"

_jcm, 2022, doi:10.3390/jcm11185477_

Round 1

Reviewer 1 Report

I think that the article is valuable in its effort to elucidate some genetics characteristic that could influence HD patients survival and that could be useful to stratify the risk in order to do appropriate intervention in the future. Nevertheless there some points that need to be clarified:

-        -  the study is presented as a “retrospective observational study” and this have to be change or more explicated: a retrospective study usually analyzes data already collected (as for clinical purpose or for other different research studies) but in this case, blood and serum samples were obtained during the study in order to test the chosen SNV and serum concentration of different protein derived from tested genes, moreover the need to collect blood samples could not be stated as fully observational because an exam other than clinical routine was needed. Please change the definition of the study (for example a cohort genetic study) or better justify the actual definition

-        -  in the “study design” section categorizing death causes only “cardiovascular” causes are mentioned. However, in the results and discussion sections and in figures many times a distinction is made between cardiovascular and cardiac mortality: please explain the difference between the two definitions in methods

-       -   in the method section please specify:

1- why are these gene chosen

2- how many SNV were analyzed,

3- what are the SNV analyzed (it would be better to clarify them here (with plain text or tables) than to stated that they are in supplementary files);

4- why they were chosen (previous studies? pre-clinical results? etc)

-        -  in the introduction please give a short description of all genes analyzed to allow readers to understand what are their many characteristics and the metabolic pathways involved; explain gene abbreviation for all genes (i.e. forxhead box protein O3 (FOXO3))

-       -   the paragraph “protein plasma concentration”: among the 86 patients randomly selected how many carry the SNVs analyzed? Could a different distribution of patients (for example two cohort with the same number of subject one with SNV and the other without) influence the results? Why or why not?

-       -   if possible add in the discussion some hypothesis that could justify the association between diabetic nephropathy and ANGPTL6 or DOCK6 as already done in lines 413 and 414 for FOXO3

Author Response

We would also like to take this opportunity to express our thanks to the reviewer for their feedback and helpful comments for correction or modification.

Comment 1:  the study is presented as a “retrospective observational study” and this have to be change or more explicated: a retrospective study usually analyzes data already collected (as for clinical purpose or for other different research studies) but in this case, blood and serum samples were obtained during the study in order to test the chosen SNV and serum concentration of different protein derived from tested genes, moreover the need to collect blood samples could not be stated as fully observational because an exam other than clinical routine was needed. Please change the definition of the study (for example a cohort genetic study) or better justify the actual definition

Answer: The blood and serum samples were collected at the beginning of the study, but the clinical data were assessed from the existing medical records. Moreover, the onset of renal replacement therapy was the beginning time point in our survival analyses, which was the main reason why we described the study as retrospective. There are differences in interpretation of what constitutes ‘intervention’ in terms of blood samples or other measurements [(2011). ENCEPP CONSIDERATIONS ON THE DEFINITION OF NON-INTERVENTIONAL TRIALS UNDER THE CURRENT LEGISLATIVE FRAMEWORK (“CLINICAL TRIALS DIRECTIVE” 2001/20/EC) Agreed by the European Network of Centres for Pharmacoepidemiology and Pharmacovigilance.]. In our study, blood samples were collected on the days when routinely measured laboratory examinations were ordered, so we considered them to be normal clinical practice. However, we changed the definition of the study to a genetic cohort study in the revised version of the manuscript for clarity.

Comment 2:  in the “study design” section categorizing death causes only “cardiovascular” causes are mentioned. However, in the results and discussion sections and in figures many times a distinction is made between cardiovascular and cardiac mortality: please explain the difference between the two definitions in methods

Answer: The causes of death were recorded and categorized as cardiac (reported as myocardial infarction, sudden cardiac death, severe arrhythmias, cardiomyopathies, or cardiac failure), vascular (reported as cerebrovascular events, cerebral stroke, or generalized atherosclerosis), infectious (reported as sepsis, pneumonia, limb necrosis, pyonephrosis, or acute abdomen with peritonitis), neoplasms, or other/unknown. Due to similar pathophysiological mechanisms contributing to diseases leading to cardiac and vascular deaths, we combined them in a broader category of cardiovascular deaths. A detailed description of the causes of death is included in the revised version of the manuscript.

Comment 3: in the method section please specify: 
1- why are these gene chosen
2- how many SNV were analyzed, 
3- what are the SNV analyzed (it would be better to clarify them here (with plain text or tables) than to stated that they are in supplementary files); 
4- why they were chosen (previous studies? pre-clinical results? etc)

Answer: We chose the genes and SNV for the study based on the literature review. Moreover, we conducted a pilot study studying the associations of FOXO3 and IGFBP3 variants with dyslipidemia and coronary artery disease in a selected group of 280 hemodialysis patients. We added this information in the methods section. Additionally, we added the table with the characteristics of the studied nucleotide variants to the main manuscript text in the revised version of the manuscript.

Comment 4: in the introduction please give a short description of all genes analyzed to allow readers to understand what are their many characteristics and the metabolic pathways involved; explain gene abbreviation for all genes (i.e. forxhead box protein O3 (FOXO3))

Answer: We added a description of all studied genes in the introduction section of the revised manuscript (lines 59 - 93).

Comment 5: the paragraph “protein plasma concentration”: among the 86 patients randomly selected how many carry the SNVs analyzed? Could a different distribution of patients (for example two cohort with the same number of subject one with SNV and the other without) influence the results? Why or why not?

Answer: We conducted an additional analysis comparing the genotype frequencies of the studied nucleotide variants in the patients in whom the plasma concentrations of the studied proteins were assessed and those in which a protein analysis was not performed. We observed no significant differences between the groups except for ANGPTL6 rs8112063 (Table S13). In the group of 86 patients with measured protein plasma levels, we observed lower frequency distribution of the rare C allele compared with the other patients (p = 0.021). This could theoretically lead to underpowered statistical tests. We added this information in the limitation section.

Comment 6: if possible add in the discussion some hypothesis that could justify the association between diabetic nephropathy and ANGPTL6 or DOCK6 as already done in lines 413 and 414 for FOXO3

Answer:  We have added additional potential mechanisms in which ANGPTL6, ANGPTL8, and DOCK6 could modulate the progression of diabetic nephropathy in the discussion section of the revised manuscript (lines 454 – 462, 486 - 494). Animal studies have shown that ANGPTL6 counteracts diet-induced obesity and insulin resistance via increasing energy expenditure (Oike, Y.; et al. Angiopoietin-Related Growth Factor Antagonizes Obesity and Insulin Resistance. Nat. Med. 2005). ANGPTL6 could therefore contribute to the development of diabetic nephropathy through pathways related to insulin resistance, which is associated with greater salt sensitivity, increased glomerular pressure, albuminuria, and kidney function decline (Adeva-Andany, M.M.; et al. Insulin Resistance Is Associated with Clinical Manifestations of Diabetic Kidney Disease (Glomerular Hyperfiltration, Albuminuria, and Kidney Function Decline). Curr. Diabetes Rev. 2022). To date, there have been no studies on putative associations between DOCK6 and diabetes. However, multiple studies revealed that the levels of ANGPTL8, whose gene is located in the corresponding intron of DOCK6  are higher in patients with diabetic nephropathy and obesity (Li, M.; et al. Association of ANGPTL8 and Resistin With Diabetic Nephropathy in Type 2 Diabetes Mellitus. Front. Endocrinol. 2021). ANGPTL8 could potentially drive the progression of DN through pathways and mechanisms related to insulin resistance or through inflammatory mechanisms (Zou, H.; et al. Circulating ANGPTL8 Levels and Risk of Kidney Function Decline: Results from the 4C Study. Cardiovasc. Diabetol. 2021; Yang, Y.; et al. Increased Circulating Angiopoietin-Like Protein 8 Levels Are Associated with Thoracic Aortic Dissection and Higher Inflammatory Conditions. Cardiovasc. Drugs Ther. 2020).

Reviewer 2 Report

The manuscript submitted by Swiderska M et al analyzes in a cross-sectional study a large cohort of prevalent patients with end-stage renal disease on hemodialysis and tested whether different SNVs are associated with patient survival and cardiovascular disease (CVD). Since mortality in patients on dialysis is partly driven by alterations of the glucose and lipid metabolisms leading to cardiovascular disease, the study was focused in 11 SNVs related with glucose and lipid metabolisms previously associated with CVD in the general populations. Additionally, they also evaluated plasma levels of different proteins related with glucose or lipid metabolisms in a randomized small sample (n=86).

General comment: The paper is well-written, the introduction and the material and methods sections are clearly explained and adequately referenced. Results are nicely presented, and the main findings of the study are appropriately discussed.

Major concerns:

11. Your cohort include only patients on hemodialysis (HD), however, no data about the dialysis modality is provided. I suggest adding some information regarding HD technique provide to your patients. For example, low flux versus high-flux HD, mixed techniques like hemodiafiltration (OL-HDF). Additionally, you can add some information about dialysis adequacy (time per week, Kt/V, vascular access, …).

22. Cardiovascular mortality in patients on HD is strongly linked to the CKD-MBD axis. Once again, no data on mineral metabolism are provided (for example, intact PTH, calcium-phosphorus product).

33. Since you conducted a cross-sectional study in a prevalent population, your study is biased. Survival analysis was conducted using as time-to-exposure event the date of HD initiation. Of course, surviving patients who started HD >15-20 years before represents a biased population since patients with lower survival are not represented. I don’t know how is distributed you cohort, but I wonder whether it is possible to do a sub analysis taking into consideration only incident patients (less than 3-6 months on HD).

Minor comments:

Plasma concentrations of FOXO3, IGFBP3, L-FABP, PCSK9, ANGPTL6, and ANGPTL8 (betatrophin) were measured in a randomly selected group of 86 HD patients using an enzyme-linked immunosorbent assay (ELISA). In the statistical analysis you should describe the method of randomization employed.

I recommend adding in the limitations that your results should be confirmed in other cohorts of Caucasian patients. Of course, I understand that it is not easy to get such cohort for the present manuscript.

Author Response

We would like to take this opportunity to express our thanks to the reviewer for their feedback and helpful comments for correction or modification. 

Comment 1: Your cohort include only patients on hemodialysis (HD), however, no data about the dialysis modality is provided. I suggest adding some information regarding HD technique provide to your patients. For example, low flux versus high-flux HD, mixed techniques like hemodiafiltration (OL-HDF). Additionally, you can add some information about dialysis adequacy (time per week, Kt/V, vascular access, …).

Answer: We extended the description regarding the HD technique in the results section of the revised manuscript. We also conducted a univariate survival analysis comparing the provided HD technique (LF-HD vs HF-HD/HDF) and observed no differences in survival probability.

Comment 2: Cardiovascular mortality in patients on HD is strongly linked to the CKD-MBD axis. Once again, no data on mineral metabolism are provided (for example, intact PTH, calcium-phosphorus product). 

Answer: We added the information on the CKD-MBD axis in Tables 2 and 3 of the manuscript. Concordantly with previous studies, we observed a significant association between Ca, phosphate, intact PTH, calcium-phosphorus product, and survival probability of HD patients. Moreover, we added intact PTH and calcium-phosphorus product to the multivariable survival analyses regarding the DOCK6 rs737337, ANGPTL6 rs8112063, and FOXO3 rs4946936 since they could be potential confounding factors in associations between the tested SNVs and mortality.

Comment 3: Since you conducted a cross-sectional study in a prevalent population, your study is biased. Survival analysis was conducted using as time-to-exposure event the date of HD initiation. Of course, surviving patients who started HD >15-20 years before represents a biased population since patients with lower survival are not represented. I don’t know how is distributed you cohort, but I wonder whether it is possible to do a sub analysis taking into consideration only incident patients (less than 3-6 months on HD). 

Answer: We conducted an additional survival analysis including only newly diagnosed end-stage renal disease patients, who recently started HD therapy. The subgroup analysis included 83 patients, who initiated HD therapy within 6 months prior to study enrollment (Table S10-S12, Figure 2). The only variants associated with mortality risk in this subgroup were the C allele of DOCK6 rs737337 and G allele of DOCK6 rs17699089. The former was associated with increased risk of cardiovascular and cardiac deaths and the latter with all-cause, cardiovascular and cardiac mortality. Due to the small size of the subgroup, we have not performed further multivariable Cox regression analyses with adjustment for clinicsl parameters to avoid bias.

Minor comments:

  1. Plasma concentrations of FOXO3, IGFBP3, L-FABP, PCSK9, ANGPTL6, and ANGPTL8 (betatrophin) were measured in a randomly selected group of 86 HD patients using an enzyme-linked immunosorbent assay (ELISA). In the statistical analysis you should describe the method of randomization employed.

Answer: We assigned selected HD patients in the protein measurement group using simple randomization. This information has been added to the methods section of the revised manuscript.

     2. I recommend adding in the limitations that your results should be confirmed in other cohorts of Caucasian patients. Of course, I understand that it is not easy to get such cohort for the present manuscript. 

Answer: We added this suggestion in the limitation section of the revised manuscript.

Round 2

Reviewer 1 Report

I thank the authors for their answers, they fully addressed the previous concerns.

Reviewer 2 Report

None. The authors have properly answered my concerns.